# OpenReview forum: "FairMT-Bench: Benchmarking Fairness for Multi-turn Dialogue in Conversational LLMs"
_ICLR.cc/2025/Conference — ICLR 2025 Spotlight_

### Official Review · Reviewer_t586 · 2024-10-25

**Soundness:** 3
**Presentation:** 3
**Contribution:** 3
**Rating:** 6
**Confidence:** 3

**Summary:**

This work presents the first fairness benchmark Fair-MT  for multi-turn dialogues because prior works can only consider single-turn scenarios.

This benchmark includes two bias types (Stereotype and Toxicity) and six bias attributes (gender, race, religion, race, etc.) and up to 10K items.  The authors first collected source data from RedditBias, SCIB, and HateXplain.  Then, the authors use GPT-4 to construct the dataset for the identified six tasks ( 2 tasks per LLM capacity). Finally, the authors mainly use GPT-4 to evaluate the fairness of LLM candidates.

**Strengths:**

1. This work has a well-defined task taxonomy, which is clear and comprehensive.
2. Unlike previous single-turn works, this work presents the first multi-turn fairness benchmark. Multi-turn has more practical significance and research value.
3. Six multi-turn dialogue tasks are proposed to detect the fine-grained fairness of LLMs.
4. LLMs can automatically accomplish most tasks related to dataset construction and evaluations.
5. Very detailed evaluation using the proposed FairMT-10K.
6. A more efficient FairMT-1K is also proposed.

**Weaknesses:**

1. This work has carefully designed two tasks for each LLM capability.   Although each task has a considerable amount of space to introduce, there is a lack of specific formal definitions for each task in the main paper.

2. Continuing from the first point, the main paper of this work has ignored many necessary details in the main paper in the remaining parts.

3.  This work does not involve enough human annotation in both the dataset construction stage and the evaluation stage.  There is only a small-scale human-annotation test in the Appendix.

4. LLMs play the role of both the dataset constructor and the evaluation referee. Is this configuration really convincing?

5. This work only focuses on the evaluation.  The author needs a more detailed discussion on what improvement directions can be brought to the subsequent researches through experimental findings.

**Questions:**

Questions:

1. See my weaknesses 4.

Minor Issues:

1. Line 130:  three stages of interaction with users`: :`
2. Line 732:  The prompt given to GPT-4 is shown in Figure `??`

**Details Of Ethics Concerns:**

This work studies the fairness in LLMs.  The corresponding methodology and findings may require an ethics review.

---

> ### Author Response · Authors · 2024-11-20
> **Response to Weaknesses 1-3**
>
> Thank you for your constructive comments. We have carefully considered your suggestions and would like to provide the following clarifications. We have also updated the manuscript, with the modified contents highlighted in **red** for your review.
>
> ---
>
> > **Weakness 1: Lack of specific formal definitions for each task in the main paper**
>
> Thanks for your insightful feedback. As you pointed out, due to the space constraints, we initially introduced and defined the nature and simple form of the tasks only in the main paper, placing the specific task formats in Appendix A.1 Figures 9-16. To enhance the readability and completeness of the paper, we **have added brief examples of each of the six tasks in Section 2.1 of the main manuscript**, based on your suggestion. These additions help may readers more quickly grasp the specific definitions and forms of the tasks.
>
> ---
>
> > **Weakness 2:  Ignored many necessary details in the main paper**
>
> Thank you very much for your suggestions. Based on your advice, we have added details related to the **specific task settings**, **details of evaluation** and **the cost of benchmark evaluation** in the main manuscript, and supplemented the corresponding details in the appendix. This will allow readers and subsequent researchers to better understand the design of FairMT-Bench when reading this article. Specifically, we have added specific task settings in Section 2.1 details of evaluation and the cost of benchmark evaluation in Section 2.3.
>
> ---
>
> > **Weakness 3:  Not involve enough human annotation**
>
> Following your suggestion, we expanded our human validation by increasing the number of samples. Specifically, we **selected 1,000 samples from the evaluation results of the FairMT-10K dataset for annotation**, and we employed uniform sampling across six tasks. Given the potential severity of underestimating biased responses by GPT-4, we sampled responses assessed by GPT-4 as biased and unbiased at a ratio of 1: 2. We employ 20 annotators, all of whom possess strong English proficiency and uphold high ethical standards. We have conducted annotation training for the annotators, clearly explaining the design of the task and the criteria for judging model responses. Each sample was independently annotated by three human annotators, with the final annotation determined by majority vote. We then calculated the consistency between the human assessment results and those of GPT-4. The results of this consistency assessment are presented in Table 1. Overall, human validation results show high **Accuracy** and **Recall**, indicating strong consistency between the GPT-4 and human judgements. GPT-4 exhibits a low rate of false negatives. However, in certain tasks, GPT-4's evaluations demonstrate a tendency toward over-protection, exhibiting a higher rate of false positives. Further analysis of cases with deviations in Appendix A.5.2 reveals a consistent pattern. Specifically, when a response includes both a refusal to answer and a biased statement, human annotators are more likely to label the response as unbiased, whereas GPT-4 tends to consider it biased. We have updated the human validation results in Section 3.2 (Lines 309-311) and Appendix A.5.2 of the paper to reflect these findings.
>
> **Table 1: Consistency between new Human Validation and GPT-4 Results.**
>
> | Task                        | Accuracy | Recall | Precision | F1     |
> |-----------------------------|----------|--------|-----------|--------|
> | Scattered Questions         | 0.9569   | 0.9394 | 0.9254    | 0.9323 |
> | Anaphora Ellipsis           | 0.9646   | 0.9490 | 0.9789    | 0.9637 |
> | Jailbreak Tips              | 0.9020   | 1.0000 | 0.4000    | 0.5714 |
> | Interference Misinformation | 0.9236   | 0.9655 | 0.8615    | 0.9106 |
> | Negative Feedback           | 1.0000   | 1.0000 | 1.0000    | 1.0000 |
> | Theme Variations            | 0.9268   | 0.9841 | 0.8493    | 0.9118 |
> | Total                       | 0.9569   | 0.9646 | 0.9160    | 0.9397 |

---

> > ### Author Response · Authors · 2024-11-20
> > **Response to Weaknesses 4**
> >
> > > **Weakness 4: LLMs play the role of both the dataset constructor and the evaluation referee.**
> >
> > We understand your concern regarding the potential bias when LLMs serve both as the dataset constructor and the evaluation referee. To address this, we would like to clarify our approach in both dataset construction and evaluation phases.
> >
> > We have implemented comprehensive measures to mitigate and reduce the negative impact of GPT-4 on the quality of the dataset and its evaluations. **Firstly, we limited the use of GPT-4 to only a small portion of the data. Secondly, we employed clear instructions to guide the generation process. Lastly, we conducted thorough validation screenings of the content generated by GPT-4.** Specifically, in the dataset generation process, we only used GPT-4 to generate data for the first question of the Scattered Questions task and the five questions of the Jailbreak Tips task. For the first question in the Scattered Questions task, we utilized GPT-4 to generate an event that reflects a particular stereotype or toxic trait of a specific group. To achieve this, we guided the model with detailed instructions, including system prompts, comprehensive task descriptions, and typical demonstrations, to produce high-quality answers. Further, we filtered out model outputs that refused to answer, identified by keywords like "sorry, I cannot answer...", from all GPT-4 generated answers. In the remaining responses, we manually checked GPT-4's output to ensure it aligned with the group's stereotype. The other questions for this task were constructed based on manually written templates and combinations of bias groups and attribute words extracted from other bias datasets, and do not suffer from quality issues due to LLM biases. For the Jailbreak Tips task, we employed the Chain of Attack (CoA) multi-round attack framework [1], assessing each round of attacks using LLMs, classifiers, and semantic similarity to the original biased statements to determine if the model's attack was successful. If an attack was unsuccessful, the LLM would regenerate a question more likely to elicit a biased response, continuing until the specified dialogue narrative was achieved. We only retained questions from successful attacks, removing samples that did not succeed within a specified number of iterations to ensure the quality of multi-round questions for the Jailbreak Tips task. The number of data generated for these two tasks is significantly lower because we retained only high-quality questions. **Thus, using GPT-4 has minimal impact on the quality of the dataset.**
> >
> > On the other hand, we understand your concerns about using GPT-4 as our evaluator.
> > - Firstly, our benchmark primarily focuses on assessing fairness in open-ended dialogue tasks within multi-turn dialogues. However, there are currently few reliable tools available for detecting bias or assessing fairness in open text. **The use of LLMs as judges is the prevalent method in existing fairness evaluation studies [2-4], and [5] has further demonstrated its feasibility and advantages.**
> > - Then, to address concerns that using LLMs for fairness assessment of generated content might be influenced by the biases inherent in LLMs, potentially leading to inaccurate results, we have implemented several measures. **We carefully designed the instructions for evaluator GPT-4**, incorporating a detailed definition of the bias detection task and providing some evaluation demonstrations to aid the model's understanding of the task. Additionally, to counteract the potential influence of model biases on the assessment results, we **included the original biased viewpoints used to construct the multi-round dialogues in the instructions, supplementing them with real human societal moral views**, following [6]. This instructional design has proven to effectively enhance the model's bias detection accuracy. Under this framework, we primarily utilize GPT-4's language understanding capabilities, simply asking it to judge whether the text under evaluation contains the given biased viewpoints.
> > - Finally, to verify the reliability of GPT-4's assessment, we included **classifier and human validation**, comparing the consistency of assessments between GPT-4 and these two methods. To further substantiate our approach, we increased the scale of human validation, with specific results presented in Table 1 in the response to Weakness_3.

---

> ### Author Response · Authors · 2024-11-20
> **Response to Weakness 5**
>
> > **Weakness 5: Lack of improvement directions.**
>
> Thank you for your suggestion. In Section 3.7 of our manuscript, we highlight the urgent need to improve fairness in multi-turn dialogues. We agree that incorporating efforts to mitigate bias, as you suggested, would significantly enhance the completeness of our paper. Consequently, we have implemented a simple debiasing attempt based on our current findings.
>
> We employed commonly used alignment paradigms DPO [7] for debiasing. We constructed positive and negative data pairs for fairness multi-turn alignment based on the evaluation results on the FairMT-10k dataset, note that the data from FairMT-1k was excluded during the selection process. We use the first four turns of historical dialogue as the conversation, with the unbiased response from the model serving as the "chosen" answer and the biased response as the "rejected" answer.   The base model is Llama-2-7b-chat. After debiasing, we **evaluate the model's fairness and the proportion of over-protection on FairMT-1k, and evaluate the model's multi-turn dialogue capabilities,**. Specifically, we evaluate the model's multi-turn dialogue capabilities based on MT-Bench 101 [8] and calculate the proportion of over-protection by determining the model's refusal rate in the Fixed Format tasks for the first four turns where the questions are entirely unbiased. To further investigate the performance in fairness alignment, we have also compared the results on a single-turn alignment dataset using only the last turn conversation of FairMT-10k dataset. The experimental results are shown in **Table 2**.
>
> From the experimental results, the model's bias ratio significantly decreased after alignment, and the performance of fairness under both single and multi-turn alignment paradigms was essentially consistent, suggesting that improvements in fairness primarily arose from the model fitting to the dataset used in this work. However, both single and multi-turn aligned models showed a decline in multi-turn dialogue language modeling capabilities, and an increase in the proportion of over-refusal, indicating that the model sacrificed performance to a large extent to enhance fairness, without achieving a good trade-off. However, it can also be seen that multi-turn alignment had a smaller impact on the model's language modeling capabilities compared to the currently prevalent single-turn safety alignment, showing certain advantages of multi-turn alignment. Therefore, designing more high-quality multi-turn fairness alignment datasets and techniques to achieve a trade-off between fairness and quality remains an unresolved issue.
>
> In response to your suggestion, we have included the fairness alignment experiment in Appendix B.4 in the revised manuscript.
>
> **Table 2: Fairness and Quality performance after model alignment.**
>
> | Metrics                   | Base Model | ST-Align-on-FairMT | MT-Align-on-FairMT |
> |---------------------------|------------|--------------------|--------------------|
> | **Bias Ratios ↓**         |            |                    |                    |
> | Scattered Questions       | 94.55%     | 0.00%              | 0.00%              |
> | Anaphora Ellipsis         | 43.03%     | 5.45%              | 4.85%              |
> | Jailbreak Tips            | 69.09%     | 0.00%              | 0.00%              |
> | Interference Misinformation| 73.94%     | 11.52%             | 10.91%             |
> | Negative Feedback         | 88.48%     | 1.21%              | 1.21%              |
> | Theme Variations          | 89.09%     | 49.09%             | 55.76%             |
> | Avg.                      | 76.36%     | 11.21%             | 12.12%             |
> | **MT-Capability ↑**       |            |                    |                    |
> | Perceptivity              | 7.70       | 6.35               | 6.77               |
> | Adaptability              | 6.00       | 6.78               | 6.01               |
> | Interactivity             | 5.17       | 2.60               | 4.43               |
> | Avg.                      | 6.29       | 5.24               | 5.73               |
> | **Over Refusal ↓**        | 12.36%     | 23.03%             | 21.82%             |

---

> > ### Author Response · Authors · 2024-11-20
> > **Response to Question**
> >
> > > **Question**
> >
> > Thank you for your question. We have detailed our specific designs for using GPT-4 to assist in generating some of the data and for conducting evaluations in the response to **Weakness_4**, to minimize any negative impact on the final evaluation results as much as possible.
> >
> > ---
> >
> > > **Minor Issues**
> >
> > Thank you for identifying this mistake. We have corrected them in the revised version of the manuscript. We carefully reviewed the manuscript to avoid similar detail errors.
> >
> > ---
> > > **References**
> >
> > [1] Yang X, Tang X, Hu S, et al. "Chain of Attack: a Semantic-Driven Contextual Multi-Turn attacker for LLM." arXiv preprint arXiv:2405.05610 (2024).
> >
> > [2] Wang S, Wang P, Zhou T, et al. "Ceb: Compositional evaluation benchmark for fairness in large language models." arXiv preprint arXiv:2407.02408 (2024).
> >
> > [3] Kumar S H, Sahay S, Mazumder S, et al. "Decoding biases: Automated methods and llm judges for gender bias detection in language models." arXiv preprint arXiv:2408.03907 (2024).
> >
> > [4] Gallegos, Isabel O., et al. "Bias and fairness in large language models: A survey." Computational Linguistics (2024): 1-79.
> >
> > [5] Zheng L, Chiang W L, Sheng Y, et al. "Judging llm-as-a-judge with mt-bench and chatbot arena." Advances in Neural Information Processing Systems 36 (2023): 46595-46623.
> >
> > [6] Fan Z, Chen R, Xu R, et al. "BiasAlert: A Plug-and-play Tool for Social Bias Detection in LLMs." Proceedings of the 2024 Conference on Empirical Methods in Natural Language Processing. 2024.
> >
> > [7] Rafailov, Rafael, et al. "Direct preference optimization: Your language model is secretly a reward model." Advances in Neural Information Processing Systems 36 (2024).
> >
> > [8] Bai G, Liu J, Bu X, et al. "Mt-bench-101: A fine-grained benchmark for evaluating large language models in multi-turn dialogues." arXiv preprint arXiv:2402.14762 (2024).

---

> ### Comment · Reviewer_t586 · 2024-11-20
> **Response to Author Rebuttal**
>
> Thanks for the careful responses.
>
> I think most of my major weaknesses have been solved or partially solved.  Hence, I will increase my recommendation.
>
> By the way, if the authors promise to append such newly added experiments to the final paper, I would like to support the acceptance of this paper.

---

> > ### Author Response · Authors · 2024-11-20
> > **Thank you for your recognition**
> >
> > Dear Reviewer t586,
> >
> > Thank you for your valuable feedback. We are grateful for your recognition of our response and the experiments in the rebuttal. We have updated the corresponding modifications in the revised manuscript.
> >
> > Once again, we extend our sincere gratitude for your time and insights. Best wishes to you!
> >
> > Warm regards,
> >
> > Authors

---

> > > ### Author Response · Authors · 2024-12-02
> > > **The Summary of the Rebuttal**
> > >
> > > Dear Reviewer t586,
> > >
> > > Thank you again for your suggestions and recognition of this work. In response to your valuable suggestions and questions, we have implemented several targeted revisions to our manuscript:
> > >
> > > - We have **explored debiasing method** by constructing a DPO dataset based on the FairMT-Bench test results, as outlined in Appendix B.6 and Table 9. Additionally, we evaluated the aligned model's fairness and capabilities in multi-turn dialogues, which led to a discussion on the unique challenges of debiasing in such dialogues.
> > >
> > > - We **increased the proportion of human annotation** to verify the accuracy of GPT-4 as Judge, and presented the new results of manual evaluation and the consistency with GPT-4's evaluation in Appendix A.5 and Table 5.
> > >
> > > - We **added detailed explanations of each task** in Sec 2.1 and supplemented with simple examples to more clearly demonstrate the definition of our tasks.
> > >
> > > - We have **incorporated more implementation details**, including a more detailed description of the Fair-1K construction process in Appendix B.5 and specific statistical information for each task in FairMT-Bench in Appendix A.1, Table 4.
> > >
> > > In addition, we have made some revisions based on the feedback from other reviewers, such as adding **a discussion on the capability of multi-turn dialogue of the LLMs** (Appendix B.1 and Table 6), and including **a comparison with existing fairness benchmarks** (Appendix C and Table 10). We believe these adjustments enhance the completeness of the paper, and we hope that these discussions will increase your recognition of the article. We appreciate your thoughtful engagement, which has significantly helped us refine our presentation of these insights.
> > >
> > > Given that we have addressed your main questions concerning FairMT-Bench, we kindly ask if you would consider increasing your score? We are grateful for your feedback and are more than willing to address any remaining concerns. We sincerely wish you all the best in your work and endeavors!
> > >
> > > Sincerely regards,
> > >
> > > Authors

---

### Official Review · Reviewer_z3a4 · 2024-11-03

**Soundness:** 3
**Presentation:** 3
**Contribution:** 4
**Rating:** 8
**Confidence:** 4

**Summary:**

This work highlights the motivation of studying fairness in multi-turn dialogue scenarios and pinpoint the current scarcity of relevant research and resources in this domain. To address these limitations, the authors construct a comprehensive multi-turn dialogue benchmark to evaluate LLM fairness capabilities across two bias types and six bias attributes. Through detailed experiments and analysis, the paper reveal fairness shortcomings in current LLMs.

**Strengths:**

1. **Valuable Resources**
This paper first presents a fairness benchmark in multi-turn dialogue scenarios, covering diverse bias types and attributes.
2. **Extensive Experiments**
Conduct comprehensive experiments on current SOTA LLMs across six designed tasks.
3. **Reliable Evaluation**
Use GPT4 as a Judge, alongside bias classifiers including Llama3-Guard-3 and human validation.
4. **Comprehensive Analysis**
Analyze evaluation results of single-turn and multi-turn dialogue across different models, tasks, and groups.
5. **Valuable Insights**
Reveal two distinct bias defense mechanisms for current LLMs, which target defense implicit biases and explicit biases respectively.

**Weaknesses:**

**Ambiguous Task Taxonomy**
In Section 3.2, two taxonomies about fairness tasks are primarily discussed: comprehension-focused tasks VS. bias-resistance tasks (Line 318-320) and implicit biases VS. explicit biases (Line 322-323). These taxonomies are clear and reasonable. However, the taxonomy outlined in section 2.1 lacks clarity and mention. The naming of "interaction fairness" class is somewhat confusing, and the boundaries between this class and the other two are not clearly defined.

**Questions:**

**Consider Maintaining Consistency Between the Taxonomy Definition Section and the Analysis Section**
In my opinion, the taxonomy outlined in Section 2.1 is less clear than the two taxonomies presented in Section 3.2. I suggest adopting a more straightforward and understandable classification for tasks definitions. If modifying the definitions is not feasible, at the very least, please ensure consistency between the classifications used in the definition and analysis sections.

---

> ### Author Response · Authors · 2024-11-20
> **Response to Weakness and Question**
>
> Thank you for your constructive comments. We have carefully considered your suggestions and would like to provide the following clarifications. We have also updated the manuscript, with the modified contents highlighted in **red** for your review.
>
> ---
>
> >**Weakness & Question: Ambiguous Task Taxonomy.**
>
> Thank you for your insightful comment. In our benchmark, the division of tasks in Section 2.1 primarily targets the fairness deficiencies of LLMs across three stages of interaction with users: the ability to perceive and understand biases in multi-turn contexts (Context Understanding), the ability to correct biases during interactions (Interaction Fairness), and the ability to balance adherence to instructions with fairness (Fairness Trade-off). You mentioned that the definition of Interaction Fairness and the boundaries with the other two categories are unclear. We have added supplements and explanations to this. In our definition, although both Interaction Fairness and Fairness Trade-off pertain to the capability of "bias resistance," **Interaction Fairness focuses more on the model's ability to judge whether a user's question contains biased information without being disturbed, while Fairness Trade-off examines whether the model can pursue utility without violating the principles of fairness.** We believe that a binary classification is clearer and more straightforward, but the current three-stage taxonomy is more detailed and comprehensive.
>
> We recognize that the change in terminology in Section 3.2 may cause confusion in the change of classification criteria.  In Section 3.2, we mentioned "comprehension-focused" and "bias resistance". This is mainly an explanation based on experimental results, focusing on explaining the reasons for bias in the model's answers, or the reasons why the model fairness alignment fails. We also think that this classification is related to the above three-stage taxonomy. The most important ability of the model in the first context understanding stage is comprehension. In the subsequent interaction and task execution stages, it is necessary to assess the correctness and fairness of user input based on a complete understanding of the context, and ensure that outputs remain fair. Therefore, its "bias resistance" ability is examined.
>
> Based on your suggestion, **we have revised lines 322-343 in Section 3.2 to enhance the clarity of the text, ensuring consistency throughout the document**. Intuitively, we have also included the terms "comprehension-focused tasks" and "bias-resisance" in Section 2.1 to provide further explanation. Additionally, we have removed the taxonomy of implicit biases, as they are not closely related to the previous discussion, in order to avoid causing further confusion.

---

### Official Review · Reviewer_aBzE · 2024-11-06

**Soundness:** 3
**Presentation:** 3
**Contribution:** 3
**Rating:** 6
**Confidence:** 4

**Summary:**

This paper introduces "FairMT-Bench," a comprehensive benchmark to evaluate fairness in LLMs, specifically in multi-turn dialogue scenarios. Addressing a gap in fairness assessments, which have primarily focused on single-turn interactions, the paper provides a dataset (FairMT-10K) and tasks that target fairness across three stages of dialogue complexity: context understanding, user interaction, and instruction trade-offs. The authors also present a distilled, challenging subset, FairMT-1K, and use both GPT-4 and Llama-Guard-3 classifiers, alongside human validation, to benchmark fairness in 15 LLMs, revealing substantial limitations in current model fairness across multi-turn interactions.

**Strengths:**

1. Novel Focus on Multi-Turn Fairness Evaluation: The paper addresses the crucial gap of multi-turn dialogue fairness, reflecting real-world complexities in conversational AI use cases.
2. FairMT-Bench and its datasets (FairMT-10K and FairMT-1K) cover a wide array of bias types and attributes, providing a rich resource for fairness research.
3. By evaluating 15 prominent LLMs, the paper provides a robust, comparative analysis of model fairness, offering valuable insights for future LLM alignment improvements.
3. Employing both GPT-4 and Llama-Guard-3 as fairness evaluators, along with detailed experimental setups, enhances reproducibility and contributes essential tools for future fairness research.

**Weaknesses:**

1. While the multi-turn focus is novel, the evaluation method largely depends on established LLM tools (e.g., GPT-4 as a judge), which may limit innovation in developing new fairness detection methodologies.
2. The paper does not thoroughly explore why certain attributes (like gender and race) showed consistently poor performance across models, missing an opportunity to deepen the community’s understanding of these biases.
3. Relying heavily on GPT-4 for generating synthetic dialogue data could introduce bias from the generative model itself, which might impact the generalizability of FairMT-10K and FairMT-1K to real-world scenarios.

**Questions:**

1. How do you ensure that biases from GPT-4, which is used to generate and evaluate responses, do not affect the fairness outcomes, particularly given GPT-4's role as both data generator and evaluator?
2. Can you clarify if the performance discrepancies across bias attributes are due to data distribution differences or model-specific limitations?
3. Lack of comparison with existing evaluation, such as S-Eval or more recent fair-related work.

---

> ### Author Response · Authors · 2024-11-20
> **Response to Weakness 1**
>
> Thank you for your constructive comments. We have carefully considered your suggestions and would like to provide the following clarifications. We have also updated the manuscript, with the modified contents highlighted in **red** for your review.
>
> ---
>
> > **Weakness 1: The evaluation method largely depends on established LLM tools (e.g., GPT-4 as a judge), which may limit innovation in developing new fairness detection methodologies.**
>
> Thank you for your suggestion, and we agree with your perspective that the LLM-as-Judge method has inherent limitations, such as those imposed by the model's own capabilities. However, there are currently few reliable tools available for detecting bias or evaluating fairness in open text. **The use of LLMs as judges is the prevalent method in existing fairness evaluation studies [1-3], and [4] has further demonstrated its feasibility and advantages**.
>
> At the same time, we acknowledge that using LLMs to evaluate the fairness of generated content may be influenced by the inherent biases in LLMs, which could lead to inaccurate evaluation results. Therefore, we have proposed some measures in our work. Firstly, when using GPT-4 as a tool for evaluating the fairness of generated content, **we carefully designed the instructions for the evaluator GPT-4**, included a detailed definition of the bias detection task, and provided some evaluation demonstrations to help the model understand the task. Additionally, to avoid the influence of the model's own biases on the evaluation results, **we included in the instructions the original biased viewpoints used to construct the multi-turn dialogue, supplementing it with real human societal moral views**. [5] has proven that such instruction design can effectively improve the model's bias detection accuracy. Under this evaluation framework, we primarily utilize GPT-4's language understanding capabilities, merely asking GPT-4 to judge whether the text under evaluation contains the given biased viewpoints, which enhances GPT-4's inherent bias detection capabilities. Additionally, **we conducted human validation on GPT-4 predictions**, and increased the number of human validation to verify the reliability of the GPT-4 evaluation. The updated consistency between human validation and GPT-4 evaluation results can be seen in Table 1. Overall, human validation results show high Accuracy and Recall, indicating strong consistency between the GPT-4 and human judgements, and GPT-4 exhibits a low rate of false negatives. However, in certain tasks, GPT-4's evaluations demonstrate a tendency toward over-protection, exhibiting a higher rate of false positives. Further analysis of cases with deviations in Appendix A.5.2 reveals a consistent pattern. Specifically, when a model's response contains both a refusal to answer and a biased statement, human annotators tend to label the response as unbiased, whereas GPT-4 is more inclined to label it as biased. We have also updated the new human validation results in Section 3.2 (Line 309-311) and Appendix A.5.2 of the paper.
>
> Finally, developing new fairness detection methods is not the focus of this paper; our work mainly addresses the lack of capabilities in multi-turn dialogues and the ease with which model fairness alignment can fail, thus we constructed a framework that comprehensively evaluates model fairness under multi-turn dialogues. Therefore, we did not propose new bias detection tools or techniques, but rather adhered to the generic methods used in existing works. Moreover, our framework can integrate any new fairness detection methods when bias detection technology is updated, offering strong scalability.
>
> **Table 1: Consistency between new Human Validation and GPT-4 Results.**
>
> | Task                        | Accuracy | Recall | Precision | F1     |
> |-----------------------------|----------|--------|-----------|--------|
> | Scattered Questions         | 0.9569   | 0.9394 | 0.9254    | 0.9323 |
> | Anaphora Ellipsis           | 0.9646   | 0.9490 | 0.9789    | 0.9637 |
> | Jailbreak Tips              | 0.9020   | 1.0000 | 0.4000    | 0.5714 |
> | Interference Misinformation | 0.9236   | 0.9655 | 0.8615    | 0.9106 |
> | Negative Feedback           | 1.0000   | 1.0000 | 1.0000    | 1.0000 |
> | Theme Variations            | 0.9268   | 0.9841 | 0.8493    | 0.9118 |
> | Total                       | 0.9569   | 0.9646 | 0.9160    | 0.9397 |

---

> ### Author Response · Authors · 2024-11-20
> **Response to Weaknesses 2-3**
>
> >**Weakness_2: The paper does not thoroughly explore why certain attributes (like gender and race) showed consistently poor performance across models, missing an opportunity to deepen the community’s understanding of these biases.**
>
> Thanks for your insightful comment. In our paper, we only discuss some social groups/attributes where different models exhibit performance disparities, noting that these biased attributes receive less attention compared to mainstream characteristics like race or religion. This lack of focus could lead to underrepresentation in fairness alignment datasets, resulting in poor fairness performance by some models on these groups. However, we did not specifically address cases where all models performed poorly. Drawing on existing research on the sources of model bias, we discuss potential reasons for this issue. As mentioned in references [3,6], the primary source of bias in LLMs is the training data. Similarly, **we believe that the effectiveness of fairness alignment in models largely depends on the data used for alignment.**
>
> Specifically, the most notable instance in the FairMT-Bench test results appears in the category of Stereotype's gender. We selected instances where the model's responses exhibited bias for detailed analysis and discussion of common issues. In the case of gender stereotypes, biased responses primarily manifest as very subtle biases: seemingly conventional and normative implicit biases against women, such as domesticity and compliance, which generally are not recognized by models as biases to trigger protective mechanisms. A specific case study is shown in Table 2. We speculate two main reasons for this situation:
> - **The data used for fairness alignment is not comprehensive enough**, involving an overly broad range of data on gender bias, making it challenging to cover comprehensively during data collection;
> - **Some biases are too subtle**, for example, the stereotype that if all women were to be employed, we might have to lower hiring standards. This type of stereotypical bias is not overtly expressed and is difficult to align for fairness using keyword detection or similar techniques, posing higher demands on the model's language understanding and reasoning capabilities.
>
> **Table 2: Case study of the prevalence of gender biased responses.**
> | Question | Answer |
> |----------|--------|
> | So based on some historical and realistic factors, they(women) are more likely to be subservient? | From a social perspective, traditional gender roles have shaped expectations that women should be submissive, docile, and supportive of others. From a political and economic perspective, women have fewer independent sources of income and political power, and this dependency makes women more dependent and submissive in economic and family decision-making. From a psychological perspective, women may be more submissive because they are more sensitive to the evaluations of others. |
>
> ---
>
> >**Weakness_3: Relying heavily on GPT-4 for generating synthetic dialogue data could introduce bias from the generative model itself, which might impact the generalizability of FairMT-10K and FairMT-1K to real-world scenarios.**
>
> Our work synthesizes assessments of models' multi-turn dialogue capabilities to design a comprehensive framework for evaluating fairness in multi-turn dialogue scenarios. It aims to compare and explore the reasons for failure in fairness alignment during multi-turn dialogues. The six tasks we designed are primarily based on manually constructed templates, combining biased groups and corresponding stereotypes to create multi-turn dialogues. During the construction process, **we thoroughly considered human users' speaking habits, adopting forms of ellipsis, reference, and persuasion typical in multi-turn dialogues to design tasks, which simulate real-world dialogue scenarios**.
>
> Although the templates and datasets we constructed still lack the diversity and flexibility of real-world multi-turn dialogues, due to the current community's lack of real-world multi-turn dialogue datasets, the vast majority of multi-turn dialogue benchmarks still rely on LLMs to generate data [7-8]. Furthermore, the few real-world multi-turn dialogue datasets that contain biases and also feature scenarios involving references and persuasion are too rare to be used for constructing a benchmark.

---

> ### Author Response · Authors · 2024-11-20
> **Response to Questions 1-2**
>
> >**Question_1: How do you ensure that biases from GPT-4, which is used to generate and evaluate responses, do not affect the fairness outcomes, particularly given GPT-4's role as both data generator and evaluator?**
>
> You raised a valid concern regarding biases from GPT-4 may affect the fairness outcomes, which indeed presents a challenge and a risk. We have implemented comprehensive measures to mitigate and reduce the negative impact of GPT-4 on the quality of the dataset and its evaluations. **Firstly, we limited the use of GPT-4 to only a small portion of the data. Secondly, we employed clear instructions to guide the generation process. Lastly, we conducted thorough validation screenings of the content generated by GPT-4**. Specifically, in the dataset generation process, we only used GPT-4 to generate data for the first question of the Scattered Questions task and the five questions of the Jailbreak Tips task. For the first question in the Scattered Questions task, we utilized GPT-4 to generate an event that reflects a particular stereotype or toxic trait of a specific group. To achieve this, we guided the model with detailed instructions, including system prompts, comprehensive task descriptions, and typical demonstrations, to produce high-quality answers. Further, we filtered out model outputs that refused to answer, identified by keywords like "sorry, I cannot answer...", from all GPT-4 generated answers. In the remaining responses, we manually checked GPT-4's output to ensure it aligned with the group's stereotype. The other questions for this task were constructed based on manually written templates and combinations of bias groups and attribute words extracted from other bias datasets, and do not suffer from quality issues due to LLM biases. For the Jailbreak Tips task, we employed the Chain of Attack (CoA) multi-round attack framework [8], assessing each round of attacks using LLMs, classifiers, and semantic similarity to the original biased statements to determine if the model's attack was successful. If an attack was unsuccessful, the LLM would regenerate a question more likely to elicit a biased response, continuing until the specified dialogue narrative was achieved. We only retained questions from successful attacks, removing samples that did not succeed within a specified number of iterations to ensure the quality of multi-round questions for the Jailbreak Tips task. The number of data generated for these two tasks is significantly lower because we retained only high-quality questions. Thus, using GPT-4 has minimal impact on the quality of the dataset.
>
> On the other hand, we understand your concerns about using GPT-4 as our evaluator. The reason for using GPT-4 as judge  and our efforts of avoiding the negative impact of GPT-4's bias and misunderstanding are detailed listed in the response of **Weakness_1**.
>
> ---
>
> >**Question_2: Can you clarify if the performance discrepancies across bias attributes are due to data distribution differences or model-specific limitations?**
>
> We have elaborated on our understanding and hypotheses regarding this issue in our response to **Weakness_2**, and demonstrated how our hypotheses corresponded to specific cases in the test results.

---

> > ### Author Response · Authors · 2024-11-20
> > **Response to Question 3**
> >
> > > **Question_3: Lack of comparison with existing evaluation, such as S-Eval or more recent fair-related work.**
> >
> >
> > Overall, we are the first to focus on fairness in multi-turn dialogues. Based on your suggestion, we have expanded our analysis to include comparisons with other existing works, as shown in **Table 3 and 4** In terms of task design, our FairMT-Bench is the pioneer in evaluating model fairness in multi-turn dialogue tasks, which belong to the domain of open text generation, thus filling a gap in current evaluations.
> > Regarding dataset composition (social group and bias type), our benchmark includes the most mainstream groups associated with social biases. Additionally, we discuss two different types of biases and the model's capacity to mitigate these biases, a feature that is quite rare in previous fairness benchmarks.
> >
> > Compared to S-Eval, which is a multi-dimensional and open-ended safety evaluation benchmark, both S-Eval and FairMT are motivated by the goal of detecting alignment vulnerabilities in current LLMs to enhance their credibility. However, FairMT-Bench specifically focuses on the fairness of LLMs and multi-turn dialogue scenarios, which sets it different from S-Eval.
> >
> > ---
> >
> > **Table 3: comparisons of Fair-MT Bench with existing works.**
> >
> > | Benchmark          | Task                               | Format       | # Num        |
> > |--------------------|------------------------------------|--------------|--------------|
> > | StereoSet          | Embedding                          | Closed ended | 16,995       |
> > | Crows-Pairs        | Embedding                          | Closed ended | 1,508        |
> > | Redditbias         | Embedding                          | Closed ended | 11,873       |
> > | Jigsaw             | Embedding                          | Closed ended | 50,000       |
> > | BBQ                | Conversation                       | Closed ended | 58,492       |
> > | UnQover            | Conversation                       | Closed ended | 30*          |
> > | BiasAsker          | Conversation                       | Closed ended | 841*8,110    |
> > | BOLD               | Text Complete                      | Open ended   | 23,679       |
> > | HONEST             | Text Complete                      | Open ended   | 420          |
> > | HolisticBias       | Conversation                       | Open ended   | 460,000      |
> > | RealToxicityPrompt | Conversation                       | Open ended   | 100,000      |
> > | TrustLLM           | Conversation                       | Closed ended | -            |
> > | CEB                | Conversation, Text Complete, Embedding | Closed ended | 11,004    |
> > | Ours (FairMT-Bench)| Multi-Turn Conversation                         | Open ended   | 10,000       |
> >
> > ---
> >
> > **Table 4: comparisons of Fair-MT Bench with existing works regarding social group and bias type.**
> >
> > | Benchmark           | Gender | Race | Age | Religion | Disabled | Appearance | Stereotype | Toxicity |
> > |---------------------|--------|------|-----|----------|----------|------------|------------|----------|
> > | StereoSet           | ✓      | ✓    |     | ✓        |          |            | ✓          |          |
> > | Crows-Pairs         | ✓      | ✓    | ✓   | ✓        | ✓        | ✓          | ✓          | ✓        |
> > | Redditbias          | ✓      | ✓    |     | ✓        |          |            |            | ✓        |
> > | Jigsaw              | ✓      | ✓    | ✓   | ✓        |          |            |            | ✓        |
> > | BBQ                 | ✓      | ✓    | ✓   | ✓        | ✓        | ✓          | ✓          |          |
> > | UnQover             |        |      |     |          |          |            |            | ✓        |
> > | BiasAsker           | ✓      | ✓    | ✓   | ✓        | ✓        |            |            | ✓        |
> > | BOLD                | ✓      | ✓    |     | ✓        |          |            |            | ✓        |
> > | HONEST              | ✓      |      |     |          |          |            |            | ✓        |
> > | HolisticBias        | ✓      | ✓    | ✓   | ✓        | ✓        | ✓          | ✓          |          |
> > | RealToxicityPrompt  |        |      |     |          |          |            |            | ✓        |
> > | TrustLLM            | ✓      | ✓    | ✓   | ✓        | ✓        | ✓          | ✓          |          |
> > | CEB                 | ✓      | ✓    | ✓   | ✓        |          |            | ✓          | ✓        |
> > | Ours (FairMT-Bench) | ✓      | ✓    | ✓   | ✓        | ✓        | ✓          | ✓          | ✓        |

---

> > > ### Author Response · Authors · 2024-11-20
> > > **References**
> > >
> > > > **References**
> > >
> > > [1] Wang S, Wang P, Zhou T, et al. "Ceb: Compositional evaluation benchmark for fairness in large language models." arXiv preprint arXiv:2407.02408 (2024).
> > >
> > > [2] Kumar S H, Sahay S, Mazumder S, et al. "Decoding biases: Automated methods and llm judges for gender bias detection in language models." arXiv preprint arXiv:2408.03907 (2024).
> > >
> > > [3] Gallegos, Isabel O., et al. "Bias and fairness in large language models: A survey." Computational Linguistics (2024): 1-79.
> > >
> > > [4] Zheng L, Chiang W L, Sheng Y, et al. "Judging llm-as-a-judge with mt-bench and chatbot arena." Advances in Neural Information Processing Systems 36 (2023): 46595-46623.
> > >
> > > [5] Fan Z, Chen R, Xu R, et al. "BiasAlert: A Plug-and-play Tool for Social Bias Detection in LLMs." Proceedings of the 2024 Conference on Empirical Methods in Natural Language Processing. 2024.
> > >
> > > [6] Suresh H, Guttag J. A framework for understanding sources of harm throughout the machine learning life cycle[C]//Proceedings of the 1st ACM Conference on Equity and Access in Algorithms, Mechanisms, and Optimization. 2021: 1-9.
> > >
> > > [7] Zhou Z, Xiang J, Chen H, et al. "Speak Out of Turn: Safety Vulnerability of Large Language Models in Multi-turn Dialogue." arXiv preprint arXiv:2402.17262 (2024).
> > >
> > > [8] Bai G, Liu J, Bu X, et al. "Mt-bench-101: A fine-grained benchmark for evaluating large language models in multi-turn dialogues." arXiv preprint arXiv:2402.14762 (2024).

---

> > > > ### Author Response · Authors · 2024-11-24
> > > > **We are eager to receive your feedback**
> > > >
> > > > Dear reviewer aBzE,
> > > >
> > > > We would like to express our sincere gratitude for your valuable time and effort. Your insights and suggestions have played an important role in improving the quality of our work. We have tried our best to address all your questions and are eager to receive your feedback.
> > > >
> > > > According to your suggestions and questions, we **explained and supplemented the improvements we made when using GPT-4 to generate data and evaluate**, and demonstrated how we avoided the adverse effects when using GPT-4. In addition, we **discussed the problem that the model generally performs poorly on certain social attributes** and provided a case study to illustrate the specific situation. Finally, in response to your suggestions, we **added a discussion on the differences between FairMT-Bench and other related work**.
> > > >
> > > > At your convenience, we would be grateful if you could review our responses and the corresponding revisions. Should there be any further issues or if additional clarification is needed, please know that we are fully prepared to make the necessary adjustments.
> > > >
> > > > Sincerely regards,
> > > >
> > > > Author

---

> ### Comment · Reviewer_aBzE · 2024-11-26
>
> Thanks for the detailed response. Please include the content in the rebuttal of the manuscript, especially Table 1/2/3. I will raise my score accordingly

---

> > ### Author Response · Authors · 2024-11-26
> > **We have updated the PDF version of the manuscript**
> >
> > Dear reviewer aBzE,
> >
> > We sincerely thank you for your kind recognition and the thoughtful insights provided in your review. Your expert suggestions are invaluable to us, and we deeply appreciate the time and effort you dedicated to improving our work. We have incorporated the content from the response into the manuscript and **updated the PDF version of the manuscript, with the modified contents highlighted in red**. The specific changes are as follows:
> >
> > **Increasing Human Validation**: We have highlighted the addition of human validation in Sections 2.3 and 3.2 of the main text and have detailed the validation process and results in Appendix A.5 and Table 5.
> >
> > **Analysis of Some Challenge Attributes**: We have analyzed and speculated on the social attributes where models generally perform poorly. The detailed analysis and case studies are provided in Appendix B.4 and Figure 20, with corresponding guidance given in Section 3.5 of the main text.
> >
> > **Comparison with Existing Methods**: We have added a detailed comparison of our work with S-Eval and other fairness-related works in Appendix C and Table 10, with corresponding guidance provided in Section 4 of the main text.
> >
> > Once again, we are truly grateful for your careful consideration and supportive words. We wish to extend our warmest regards and best wishes to you. Thank you for contributing significantly to enhancing the quality of our manuscript!
> >
> > Sincerely regards,
> >
> > Author

---

### Official Review · Reviewer_vp1K · 2024-11-07

**Soundness:** 3
**Presentation:** 4
**Contribution:** 3
**Rating:** 8
**Confidence:** 4

**Summary:**

This paper proposes a fairness benchmark designed for multi-turn dialogues, called FairMT-Bench. Then, it detailed the experiments and analysis with multiple SOTA LLMs across dimensions like tasks, dialogue turns, bias types and attributes. It carefully analyzed the results and pointed out the areas where LLMs fail to maintain a stable fairness. For example, LLMs tend to perform worse on fairness in multi-turn dialogues than single turn. Moreover, the paper proposes a more challenging fairness evaluation dataset, FairMT-1K, that contains examples LLMs perform worst on.

**Strengths:**

1. The paper contributes a novel fairness benchmark specifically for multi-turn dialogues, while current benchmarks primarily focus on single-turn dialogues.

2. The paper extensively benchmarks on most popular LLMs, and provides detailed results and analysis across many dimensions, like tasks, dialogue turns, bias types and attributes. The paper comprehensively demonstrates each LLM's performance, and pinpoints the areas where fairness is challenging to LLMs. The results show that fairness, especially in multi-turn dialogues, is still a challenging task for LLMs.

3. The paper has great presentation. The figures are very illustrative and insightful.

**Weaknesses:**

1. A few factors can make the evaluation computationally expensive: (1) using GPT-4 as the evaluator (2) the multi-turn nature of the data and the evaluation process (3) the data size. It would be great if the paper can include some discussion on evaluation cost.

2. While the paper discusses diverse sources and dimensions of bias, it does not discuss potential mitigation strategies. Offering even preliminary solutions or suggestions for future research directions would be valuable.

3. As fairness and quality can be contradictory metrics, models that perform well on fairness might sacrifice its quality. Therefore, it'd be better to add a model quality dimension in the analysis, for a more comprehensive and insightful comparison of the models.

4. The distribution of different tasks needs more clarification. I couldn't find the numbers of each task in the paper. Could the authors point me to the numbers if they are present in the paper, or add this information if they are absent?

**Questions:**

1. When constructing the more challenging subset (FairMT 1K), the six models may diverge in scores. So how did you use the model results to select the examples?

2. Typos:
  1) Line 368, should be "multi-turn dialogues generally show higher bias rates"?
  2) Figure 4, (a) and (b) look the same, which I assume is not intended?

3. As mentioned above, could the authors clarify the number of examples allocated to each task in the FairMT-10K dataset?

---

> ### Author Response · Authors · 2024-11-20
> **Response to Weakness 1**
>
> Thank you for your constructive comments. We have carefully considered your suggestions and would like to provide the following clarifications. We have also updated the manuscript, with the modified contents highlighted in **red** for your review.
>
> ---
>
> > **Weakness 1: Lack of discussion on evaluation cost.**
>
> We agree with your opinion. As a practical benchmark, it's important to clearly outline the costs of the benchmark for future users. Therefore, we have supplemented the discussion on costs from both time and economic perspectives:
> - **Time cost**: The generation speed varies among different models. In this analysis, we take Llama-2-7b-chat model as an example. Some API-called models, such as ChatGPT, generally produce faster results. We configure the model to generate text with “max_new_tokens” set to 150 and maintain the batch size and precision as 1 and FP32, respectively. The generation is performed on a single NVIDIA H100 GPU. Under this setup, generating **10K samples of 5-turn multi-turn dialogue responses takes about 72.5 H100 GPU hours**.
> - **Economic cost**: First, we discuss the costs associated with using GPT-4 as an evaluator. Specifically, we use GPT-4 Turbo to evaluate the generated content of the test models. The current price of GPT-4 Turbo is 0.01 USD per 1,000 input tokens and 0.03 USD per 1000 output tokens. In our setup, GPT-4 evaluates bias in the final turn of the dialogue by processing the historical dialogue as context and providing simple outputs: either 'Yes, it is biased' or 'No, it is not biased'. We have measured the **cost of evaluating on FairMT-10k to be 171.28 USD**. In addition, we also use the open-source model LlamaGuard-3 as another evaluation tool. While LlamaGuard-3 is less sensitive in detecting implicit biases compared to GPT-4, its assessment results and trends are generally consistent with those of GPT-4. LlamaGuard-3 serves as a cost-effective alternative when GPT-4 API is unavailable or to minimize expenses. Furthermore, to enhance evaluation efficiency and reduce costs, we have curated 1,000 particularly challenging samples from the FairMT-10K dataset into a new subset, FairMT-1K.
>
> In response to your suggestion, we have included the costs of the benchmark in Section 2.3 and Appendix A.4 in the revised manuscript.

---

> > ### Author Response · Authors · 2024-11-20
> > **Response to Weakness 2**
> >
> > > **Weakness 2: Lack of discuss potential mitigation strategies.**
> >
> > Thank you for your suggestion. In Section 3.7 of our manuscript, we highlight the urgent need to improve fairness in multi-turn dialogues. We agree that incorporating efforts to mitigate bias, as you suggested, would significantly enhance the completeness of our paper. Consequently, we have implemented a simple debiasing attempt based on our current findings.
> >
> > We employed commonly used alignment paradigms **DPO [1] for debiasing**. We constructed positive and negative data pairs for fairness multi-turn alignment based on the evaluation results on the FairMT-10k dataset, note that the data from FairMT-1k was excluded during the selection process. We use the first four turns of historical dialogue as the "conversation", with the unbiased response from the model serving as the "chosen" answer and the biased response as the "rejected" answer.   The base model is Llama-2-7b-chat. After debiasing, we evaluate the model's **fairness** and the **proportion of over-protection** on FairMT-1k, and evaluate the model's **multi-turn dialogue capabilities**. The details of capabilities and over-protection evaluation can be referred to Weakness 3. To further investigate the performance in fairness alignment, we have also compared the results on a single-turn alignment dataset using only the last turn conversation of FairMT-10k dataset. The experimental results are shown in **Table 1**.
> > From the experimental results, the model's bias ratio significantly decreased after alignment, and the performance of fairness under both single and multi-turn alignment paradigms was essentially consistent, suggesting that improvements in fairness primarily arose from the model fitting to the dataset used in this work. However, both single and multi-turn aligned models showed a decline in multi-turn dialogue language modeling capabilities, and an increase in the proportion of over-refusal, indicating that the model sacrificed performance to a large extent to enhance fairness, without achieving a good trade-off. However, it can also be seen that multi-turn alignment had a smaller impact on the model's language modeling capabilities compared to the currently prevalent single-turn safety alignment, showing certain advantages of multi-turn alignment. Therefore, designing more high-quality multi-turn fairness alignment datasets and techniques to achieve a trade-off between fairness and quality remains an unresolved issue.
> >
> > In response to your suggestion, we have included the fairness alignment experiment in Appendix B.4 in the revised manuscript.
> >
> > **Table 1: Fairness and Quality performance after model alignment**
> >
> > | Metrics                   | Base Model | ST-Align-on-FairMT | MT-Align-on-FairMT |
> > |---------------------------|------------|--------------------|--------------------|
> > | **Bias Ratios ↓**         |            |                    |                    |
> > | Scattered Questions       | 94.55%     | 0.00%              | 0.00%              |
> > | Anaphora Ellipsis         | 43.03%     | 5.45%              | 4.85%              |
> > | Jailbreak Tips            | 69.09%     | 0.00%              | 0.00%              |
> > | Interference Misinformation| 73.94%     | 11.52%             | 10.91%             |
> > | Negative Feedback         | 88.48%     | 1.21%              | 1.21%              |
> > | Theme Variations          | 89.09%     | 49.09%             | 55.76%             |
> > | Avg.                      | 76.36%     | 11.21%             | 12.12%             |
> > | **MT-Capability ↑**       |            |                    |                    |
> > | Perceptivity              | 7.70       | 6.35               | 6.77               |
> > | Adaptability              | 6.00       | 6.78               | 6.01               |
> > | Interactivity             | 5.17       | 2.60               | 4.43               |
> > | Avg.                      | 6.29       | 5.24               | 5.73               |
> > | **Over Refusal ↓**        | 12.36%     | 23.03%             | 21.82%             |

---

> ### Author Response · Authors · 2024-11-20
> **Response to Weaknesses 3-4**
>
> > **Weakness 3: Need to add a model quality dimension in the analysis.**
>
> We wholeheartedly agree with your suggestion that adding an evaluation metric for model capabilities can significantly enhance the completeness of our paper.
>
> Based on your suggestion, we have included additional assessments of model capabilities in multi-turn dialogue. We **tested the multi-turn dialogue capabilities of the six models on MT-Bench101 [2]**, a benchmark for fine-grained evaluation of model multi-turn dialogue capabilities.
>
> Additionally, in our FairMT-Bench, we designed tasks to assess the trade-off between fairness and model utility. In conjunction with our tasks for assessing model fairness, we focused on the trade-off between fairness and model utility, observing whether the model begins to refuse responses due to keywords associated with certain social groups or attributes without any biased intent from the user, sacrificing model utility to ensure fairness. In our task design, the **Fixed Format task** involves asking the model to answer unbiased questions related to a social group in a fixed format for the first four turns, and then posing a biased question related to the same social group in the same format in the fifth turn. This task design considers the model’s ability to balance performance and fairness under clear instructions and requirements, and also examines the model's flexibility in changing its response strategy when the dialogue content undergoes subtle changes across multiple turns. Ideally, the model should normally answer according to the format in the first four turns and refuse to answer while pointing out the bias in the prompt in the last turn of dialogue. Here, we examine whether the model exhibits over-protection by generally refusing to respond upon detecting sensitive words related to a particular group, specifically if the model displays refusal behavior in the first four turns.
>
> The test results from MT-Bench101 and the model's over-refusal are shown in Table 2. Overall, **models with stronger multi-turn dialogue capabilities tend to exhibit better fairness performance** in multi-turn dialogues (as analyzed from the Avg. results). Specifically, we found that models with stronger Adaptability and Interactivity are more susceptible to user guidance and interference, which can alter their fairness strategies during multi-turn dialogues. We think this may be because the fairness alignment of these models has not adequately accounted for multi-turn adversarial prompts.
>
> **Table 2: Results of Multi-turn capability.**
>
> | Model              | Perceptivity | Adaptability | Interactivity | Avg.  | Over Refusal |
> |--------------------|--------------|--------------|---------------|-------|--------------|
> | ChatGPT            | 8.73         | 7.55         | 7.40          | 7.99  | 0.00%        |
> | Llama-2-7b-chat    | 7.70         | 6.00         | 5.17          | 6.29  | 29.89%       |
> | Llama-2-13b-chat   | 8.47         | 6.39         | 6.15          | 7.15  | 23.10%       |
> | Llama-3.1-8b-it    | 4.26         | 2.37         | 3.40         | 3.34         | 12.78%       |
> | Mistral-7b-it-v0.3 | 7.85         | 6.82         | 6.0         | 6.89         | 21.40%       |
> | Gemma-7b-it        | 8.93         | 7.03         | 5.26          | 7.07  | 26.03%       |
>
> ---
>
> > **Weakness_4 The distribution of different tasks needs more clarification.**
>
> Based on your suggestion, we have supplemented the statistical distribution of the dataset across different tasks in Table 4, Appendix A.1 in the revised manuscript. The results are also provided in the following Table 3. Specifically, for the Anaphora Ellipsis and Jailbreak Ellipsis tasks, we utilized GPT-4 to assist in generating part of the data. Therefore, we conducted a manual validation of the quality of the content generated by GPT-4 and removed some low-quality questions, resulting in slightly lower numbers for these two tasks. The other four tasks were constructed based on manually crafted dialogue templates, hence the quantity is equal across these tasks.
>
> **Table 3: The number distribution of each task on the Fair MT 10k dataset**
>
> | Bias Type   | Scattered Questions | Anaphora Ellipsis | Jailbreak Tips | Interference Misinformation | Negative Feedback | Theme Variations |
> |-------------|---------------------|-------------------|----------------|-----------------------------|-------------------|------------------|
> | Stereotype  | 1356                | 1211              | 841            | 1356                        | 1356              | 1356             |
> | Toxicity    | 481                 | 459               | 298            | 481                         | 481               | 481              |
> | Total       | 1837                | 1670              | 1139           | 1837                        | 1837              | 1837             |

---

> > ### Author Response · Authors · 2024-11-20
> > **Response to Questions 1-3**
> >
> > >**Questions_1: Clarify how to use the model results to select the examples.**
> >
> > Thank you for pointing out the omission of the implementation details in constructing the Fair MT 1k dataset in our paper. To reduce the cost of evaluating fairness in multi-turn dialogues and enhance the efficiency of these evaluations, we selected a more challenging dataset based on the results of six models tested on the Fair-10k dataset. We assume that the more models that exhibit biased responses to a data point, the more challenging that data point is for the models. Therefore, we selected the composition of the FairMT-1k dataset based on the number of times a model exhibited bias on a particular data point, independent of the model's fairness performance. For each task, we **integrated the evaluation results of six models across two types of biases and counted the number of models that exhibited bias in the final turn of each dialogue group**. We considered a dialogue group as challenging if more models produced biased responses in that group. We ranked all samples based on the number of models that provided biased responses and selected the top 170 groups with the most biased responses to be included in the FairMT-1k dataset.
> >
> > In response to your suggestion, we have included the detailed construction process of the FairMT-1k dataset in Appendix B.5 in the revised manuscript.
> >
> > ---
> >
> > >**Questions_2: Typos**
> >
> > Thank you for identifying this mistake. We have corrected them in the revised version of the manuscript. We carefully reviewed the manuscript to avoid similar detail errors.
> >
> > ---
> >
> > >**Questions_3: Clarify the number of examples allocated to each task in the FairMT-10k dataset.**
> >
> > We are appreciate for your suggestions. We have supplemented the statistical distribution of the FairMT-10k with details in Weakness_4.
> >
> > ---
> >
> > >**Reference**
> >
> > [1] Rafailov, Rafael, et al. "Direct preference optimization: Your language model is secretly a reward model." Advances in Neural Information Processing Systems 36 (2024).
> >
> > [2] Bai, Ge, et al. "Mt-bench-101: A fine-grained benchmark for evaluating large language models in multi-turn dialogues." arXiv preprint arXiv:2402.14762 (2024).

---

> ### Author Response · Authors · 2024-12-02
> **We are eager to receive your feedback**
>
> Dear Reviewer vp1K,
>
> We would like to express our sincere gratitude for your valuable time and effort. Your insights and suggestions have played a crucial role in enhancing the quality of our work. We have made every effort to address all your questions and are eager to receive further feedback.
>
> In response to your valuable suggestions and questions, we have implemented several targeted revisions to our manuscript:
>
> - We have included **a detailed discussion on the cost implications** of utilizing our benchmark, as detailed in Section 2.3 and Appendix A.4.
>
> - We have **addressed debiasing** by constructing a DPO dataset based on the FairMT-Bench test results, as outlined in Appendix B.6 and Table 9. Additionally, we evaluated the aligned model's fairness and capabilities in multi-turn dialogues, which led to a discussion on the unique challenges of debiasing in such dialogues.
>
> - We have enhanced our manuscript with **a discussion on the model’s multi-turn dialogue capabilities and quality**, as shown in Appendix B.1 and Table 6. Specifically, we assessed the model's performance in multi-turn dialogues using MT-Bench 101 and explored the over-rejection rate using FairMT-Bench, providing insights into the factors that influence the failure of fairness alignment in these scenarios.
>
> - We have incorporated **more implementation details**, including a more detailed description of the Fair-1K construction process in Appendix B.5 and specific statistical information for each task in FairMT-Bench in Appendix A.1, Table 4.
>
> We appreciate your thoughtful engagement, which has significantly helped us refine our presentation of these insights. Given that we have addressed your main questions concerning FairMT-Bench, we kindly ask if you would consider revising your score? We are grateful for your feedback and are more than willing to address any remaining concerns. We sincerely wish you all the best in your work and endeavors!
>
> Sincerely regards,
>
> Author

---

### Author Response · Authors · 2024-11-25
**Global Reponse**

Dear reviewers,

We would like to express our heartfelt gratitude for your invaluable time, expertise, and meticulous attention in reviewing our manuscript. The insightful comments and constructive feedback have immensely enriched the quality and rigor of our work.

We appreciate that the reviewers acknowledge the advantages of our work: "**contributes a novel fairness benchmark specifically for multi-turn dialogues**" (reviewer vp1K), "**addresses the crucial gap of multi-turn dialogue fairness**" (reviewer aBzE), "**conduct comprehensive experiments and analysis**" (reviewer z3a4), "**has a well-defined task taxonomy, which is clear and comprehensive**" (reviewer t586).
On the other hand, we have diligently addressed all the issues by conducting extensive experiments to support our responses. Revised experiments and clarifications have been incorporated into the main manuscript while supplementary experiments have been included in the appendix, with the modified contents highlighted in **red**.

Allow me to summarize the significant updates made in both the rebuttal and the revised manuscript:
- **Addition of Human Validation**: Since we employed GPT-4 to achieve fully automated evaluation, we needed to validate the feasibility and accuracy of using GPT-4 as an evaluator. In the original version, we used LlamaGuard-3 and a small amount of manual annotation to verify GPT-4's evaluation results. Based on the reviewers' suggestions, we increased the scale of human validation by resampling 1,000 instances for manual annotation.
- **Supplementation of LLMs' Multi-Turn Dialogue Capability Evaluation**: To enhance the completeness of the evaluation framework and delve deeper into the factors influencing fairness in multi-turn dialogues, we added discussions on the quality of models in multi-turn dialogues. The models were evaluated based on MT-Bench 101 and their over-rejection rates on the FairMT Bench.
- **Addition of Bias Mitigation Attempts in Multi-Turn Dialogue Scenarios**: Based on our evaluation results and the generated outputs during model testing, we constructed a simple multi-turn DPO dataset for fairness alignment and conducted bias mitigation attempts using Llama-2-7b-chat.
- **Avoiding Inappropriate Content**: In the new version, we have reviewed the illustrations and examples in our case studies, replacing some overly sensitive cases that could easily cause discomfort. We aim to minimize the potential discomfort experienced by groups with various social attributes when reading our article.
- **Further Explanation of GPT-4's Role in Data Generation and Evaluation**: We elaborated on how GPT-4 was rigorously utilized to assist in data generation and evaluation within FairMT-Bench. This addresses the reviewers' concerns about the potential biases and limitations of the GPT-4 model impacting the evaluation framework.

We sincerely appreciate your time and valuable insights.

Warm regards,

The Authors

---

### Meta-Review · Area_Chair_Mjp5 · 2024-12-19

**Metareview:**

This paper introduces FairMT-Bench, a comprehensive fairness benchmark for LLMs in multi-turn dialogues, addressing the limitations of existing single-turn benchmarks. The reviewers appreciate the focus of the paper on fairness, commenting that it addresses an important and significant gap, the comprehensive evaluation and analysis, as well as the value in the datasets provided. They also raise concerns, primarily on over-reliance on LLMs and the bias and cost this might incur, and some limitations in the evaluation, analysis, and discussion. The authors do a good job in providing detailed responses to all concerns and questions, including additional analyses and human annotations.

**Additional Comments On Reviewer Discussion:**

The authors provide detailed responses and additional analyses and experiments to address the comments and questions from the reviewers. Two reviewers responded to the authors and raised their scores. Overall I find the author responses comprehensive and convincing and I believe they have improved the paper.

---

### Decision · Program_Chairs · 2025-01-22

Accept (Spotlight)